# Paeoniflorin Enhances Endometrial Receptivity through Leukemia Inhibitory Factor

**DOI:** 10.3390/biom11030439

**Published:** 2021-03-16

**Authors:** Hye-Rin Park, Hee-Jung Choi, Bo-Sung Kim, Tae-Wook Chung, Keuk-Jun Kim, Jong-Kil Joo, Dongryeol Ryu, Sung-Jin Bae, Ki-Tae Ha

**Affiliations:** 1Korean Medical Research Center for Healthy Aging, Pusan National University, Yangsan, Gyeongsangnam-do 50612, Korea; rin8998@pusan.ac.kr (H.-R.P.); hjchoi@pusan.ac.kr (H.-J.C.); kkc0704@pusan.ac.kr (B.-S.K.); twchung@pusan.ac.kr (T.-W.C.); Dr.NowOrNever@pusan.ac.kr (S.-J.B.); 2Department of Korean Medical Science, School of Korean Medicine, Pusan National University, Yangsan, Gyeongsangnam-do 50612, Korea; 3Department of Clinical Pathology, Daekyeung University, Gyeongsan, Gyeongsanabuk-do 38547, Korea; biomed@tk.ac.kr; 4Department of Obstetrics and Gynecology, School of Medicine, Pusan National University Hospital, Busan 49241, Korea; jkjoo@pusan.ac.kr; 5Department of Molecular Cell Biology, Sungkyunkwan University School of Medicine, Suwon 16419, Korea; freefall@skku.edu

**Keywords:** paeoniflorin, endometrial receptivity, embryo implantation, leukemia inhibitory factor

## Abstract

Despite advances in assisted reproductive technology, treatment for deficient endometrial receptivity is a major clinical unmet need. In our previous study, the water extract of *Paeonia lactiflora* Pall. enhanced endometrial receptivity in vitro and in vivo via induction of leukemia inhibitory factor (LIF), an interleukin (IL)-6 family cytokine. In the present study, we found that paeoniflorin, a monoterpene glycoside, is the major active compound of *P. lactiflora*. Paeoniflorin significantly improved the embryo implantation rate in a murine model of mifepristone (RU486)-induced implantation failure. In addition, paeoniflorin increased the adhesion of human trophectoderm-derived JAr cells to endometrial Ishikawa cells through the expression of LIF in vitro. Moreover, using the National Center for Biotechnology Information (NCBI) Gene Expression Omnibus (GEO) database of the human endometrium, we confirmed that LIF signaling is a key regulator for improving human endometrial receptivity. Therefore, these results suggest that paeoniflorin might be a potent drug candidate for the treatment of endometrial implantation failure by enhancing endometrial receptivity.

## 1. Introduction

A receptive endometrium is a crucial factor for successful pregnancy [1]. Despite advances in assisted reproductive technology (ART), the implantation success rates of transferred embryo have not sufficiently improved [2]. To evaluate and improve endometrial receptivity, vigorous studies, including on endometrial receptivity array and intentional endometrial injury, have been conducted [3,4]. However, there are very limited clinical options for improving endometrial receptivity [5]. Thus, more precise, comprehensive, and novel approaches for enhancing endometrial receptivity are needed.

Endometrial receptivity is regulated by diverse factors such as hormones, cytokines, and growth factors [1,3]. Among these factors, leukemia inhibitory factor (LIF), a cytokine belonging to the interleukin (IL)-6 family, has been regarded as a key player regulating uterine receptivity [6,7]. LIF exhibits a multifaceted action by activating many pathways, including epithelial–mesenchymal transition, angiogenesis, stromal cell decidualization, cell growth, and integrin signaling [6]. Previous studies have demonstrated that LIF enhances endometrial receptivity by activating Arg–Gly–Asp (RGD)-binding integrins such as integrin αV, β3, and β5 [8,9,10]. Thus, in this study, we aimed to identify novel agents that improve endometrial receptivity by inducing LIF expression [11,12,13,14].

In our previous study, the root of *Paeonia lactiflora* Pall. enhanced endometrial via induction of LIF [12]. *P. lactiflora* is widely used for the treatment of gynaecological problems, pain, cramp, and giddiness in traditional Chinese medicine [15]. It has also been studied for its effects on diverse diseases including tumors, hepatitis, diabetes, arthritis, and colitis [16,17]. According to previous reports, paeoniflorin, a major ingredient of *P. lactiflora*, has anti-inflammatory effects on various inflammatory disorders including inflammatory bowel disease, rheumatoid arthritis, asthma, and psoriasis [18]. In addition, paeoniflorin has been reported to suppress epithelial–mesenchymal transition in mouse pulmonary fibrosis [19] and attenuate insulin resistance and hepatic steatosis [20]. Although the effect of *P. lactiflora* on endometrial receptivity was previously demonstrated by the authors [12,21], no study has reported the effect of paeoniflorin on embryo implantation.

In this study, we demonstrated that paeoniflorin increased embryo implantation in both in vitro and in vivo models. In addition, we found that the expression of LIF played an important role in paeoniflorin-stimulated adhesion of trophoblasts to the endometrium. Therefore, our findings present evidence showing that paeoniflorin could be a potent agent for improving endometrial receptivity.

## 2. Materials and Methods

### 2.1. Materials

Paeoniflorin (≥98% purity using high-performance liquid chromatography (HPLC)), gallic acid (≥99% purity, HPLC), (±)-catechin (≥96% purity, HPLC), (+)-catechin hydrate (≥98% purity, HPLC), methyl gallate (≥90% purity, HPLC), paeonol (≥98% purity, HPLC), mifepristone (RU486; RU, progesterone receptor antagonist), 3-(4,5-dimethylthiazol-2-yl)-2,5-diphenyltetrazolium bromide (MTT), and anti-β-actin antibody were purchased from Sigma-Aldrich (St. Louis, MO, USA). Antibodies against LIF were supplied by Santa Cruz Biotechnology Inc. (Santa Cruz, CA, USA).

### 2.2. Traditional Chinese Medicine Systems Pharmacology (TCMSP) Analysis

The Traditional Chinese Medicine Systems Pharmacology TCMSP (https://tcmspw.com/ (accessed on 15 March 2021)) database is useful for evaluating the absorption, distribution, metabolism, and excretion (ADME) processes of compounds. To screen *P. lactiflora* for potential candidate components, we selected compounds with an oral bioavailability (OB) ≥30% and drug likeness (DL) ≥0.18.

### 2.3. Animals

Male and female C57BL/6 mice (7–8 weeks old, weighing 20–22 g) inbred in a specific pathogen-free facility were purchased from Orient Bio, Co. (Seongnam, Korea). They were housed separately and had free access to water and a standard diet on a 12 h light/dark cycle. All experimental procedures were reviewed and approved by the Animal Research Ethics Committee of Pusan University of Korea (no. PNU-2017-1606).

### 2.4. Animal Models and Treatment

The experiment using the embryo implantation failure model was performed as previously described [12]. Briefly, 40 female mice were randomly divided into four groups: control, RU, paeoniflorin + RU, and paeoniflorin groups. Female mice in the paeoniflorin + RU and paeoniflorin groups were orally administered paeoniflorin (8 mg·kg^−1^·day^−1^) using oral gavage needles. Seven days after paeoniflorin treatment, all the female mice were caged with males (ratio, 1:1) at 6:00 p.m., and day 1 of pregnancy was defined by the presence of vaginal plugs the following morning. Female mice in the RU and paeoniflorin + RU groups were injected subcutaneously with 0.08 mg/0.1 mL RU solution on day 4–7 of pregnancy at 9:00 a.m.. Seven days after RU treatment, all mice were euthanized, and both uterine horns were excised to determine the number of implantation sites. The embryos implanted on each uterine site were counted.

### 2.5. Cell Culture

Human endometrial Ishikawa cells provided by Dr. Jacques Simard (CHUL Research Center, Quebec City, QC, Canada) and trophoblastic JAr cells purchased from the Korean Cell Line Bank (Seoul, Korea) were incubated with 10% heat-inactivated fetal bovine serum (FBS; Thermo Fisher Scientific, Waltham, MA, USA) in Dulbecco’s modified Eagle’s medium (DMEM; Welgene, Daegu, Korea) and Roswell Park Memorial Institute (RPMI) 1640 (Welgene), respectively. The cultures were maintained at 37 °C in a humidified incubator containing 5% carbon dioxide (CO_2_).

### 2.6. Cell Viability Assay

The MTT assay was performed to measure the in vitro cytotoxicity of paeoniflorin on Ishikawa cells [22]. The cells were cultured in 24-well plates, treated with the indicated concentrations of paeoniflorin for 24 h, and washed three times with phosphate-buffered saline (PBS); then, MTT solution (0.5 mg/mL) was added to each well. After a 4 h incubation, the cells were dissolved in a solution of dimethyl sulfoxide (DMSO)/ethanol (EtOH, 1:1) and the absorbance was measured at 540 nm using a microplate reader (SpectraMax M2; Molecular Devices, San Jose, CA, USA). The proportion of live cells was determined and expressed as a percentage.

### 2.7. Western Blot Analysis

Western blot assays were performed as described previously [23]. Ishikawa cells were collected by scraping in 1% NP-40 lysis buffer and incubated for 1 h at 4 °C. The cell lysates were clarified by centrifuge at 15,000 rpm for 30 min, followed by protein quantification with Bradford assay (Bio-Rad, # 5000006). The extracted proteins (20 μg) were separated using sodium dodecyl sulfate polyacrylamide gel electrophoresis (SDS-PAGE) and transferred onto nitrocellulose membranes (Amersham Protran, 10600003). After preincubation with blocking solution (5% skim-milk/Tris-buffered saline plus Tween (TBS-T)) for 1 h, the membranes were incubated with specific primary antibodies, which were diluted to 1:1000 in TBS-T, at 4 °C overnight. Then, the membranes were incubated with horseradish peroxidase-conjugated secondary antibodies (1:1000 in TBS-T), and the signals were visualized using an enhanced chemiluminescence (ECL) system (GE Healthcare, Chicago, IL, USA).

### 2.8. Reverse Transcription Polymerase Chain Reaction (RT-PCR)

Total RNA from each sample was isolated using RiboEx^TM^ (GeneAll, Seoul, Korea) and then reverse-transcribed using oligo-dT primers with M-MLV reverse transcriptase (Enzynomics, Daejeon, Korea). The complementary DNA (cDNA) was amplified by PCR using AccuPower^®^ PCR PreMix (Bioneer Co., Daejeon, Korea). The primers used were as follows: LIF forward, 5′–GGCCCGGACACCCATAGACG–3′ and LIF reverse, 5′–CCACGCGCCATCCAGGTAAA–3′; β-actin forward, 5′–CAAGAGATGGCCACGGCTGCT–3′ and β-actin reverse, 5′–TCCTTCTGCATCCTGTCGGCA–3′.

### 2.9. Cell Adhesion Assay

Cell adhesion assays were performed as described previously with minor modification [24]. Ishikawa, pLKO.1, and siLIF-transfected cells were cultured in six-well plates for 24 h and then treated with paeoniflorin for 48 h. The JAr cells were labeled with 5-chloromethylfluoresceindiacetate (CMFDA) fluorescence dye (CellTracker Green; Life Technologies) for 15 min at 37 °C. The labeled JAr cells were then washed three times with PBS and gently added to Ishikawa cell monolayers. After gentle shaking at 40 rpm for 30 min at 37 °C, the cells were washed three times to remove non-adherent JAr cells. The number of attached JAr cells was visualized using a fluorescent microscope (Axio Imager M1, Zeiss, Oberkochen, Germany) and calculated using the ImageJ software (National Institutes of Health (NIH), Bethesda, MD, USA).

### 2.10. Knockdown of LIF Expression

To temporally reduce the expression of LIF, we purchased two different small interfering RNAs (siRNAs, Bioneer Co.) and tested the efficiency of knockdown of human LIF expression. The siRNA sequences used in this study were as follows: siLIF#1 sense, 5′–CAGAUGUUCCUGCCUUAGA–3′ and siLIF#1 antisense, 5′–UCUAAGGCAGGAACAUCUG–3′; siLIF#2 sense, 5′–CCUCCGACAAGAUGAUGGU–3′ and siLIF#2 antisense, 5′–ACCAUCAUCUUGUCGGAGG–3′. Two siRNAs (0.8 nM) were transfected into Ishikawa cells using Lipofectamine 2000 (Invitrogen, Carlsbad, CA, USA). After 6 h, the medium was changed to complete growth medium. A non-targeting siRNA (SN-1001-CFG; Bioneer Co.), with low homology to human genomic DNA was used as a negative control. The efficiency of the siRNAs was evaluated using RT-PCR.

### 2.11. Bioinformatic Analysis

Publicly available microarray data were used for transcript expression analysis [25,26,27]. Transcriptomic data for the human endometrium obtained from patients with recurrent implantation failure (RIF) and fertile women (accession number GSE71835, GSE92324, and GSE26787) and data for normal tissue from various phases of the menstrual cycle (accession number GSE4888) were analyzed using the Hallmark and gene set enrichment analysis (GSEA) [28].

### 2.12. Protein–Protein Physical Interaction Network Analysis

The networks were created using GeneMANIA (https://genemania.org (accessed on 15 March 2021)), and the physical interaction between two proteins was represented using an edge [29].

### 2.13. Statistical Analysis

The intensity of the bands obtained from RT-PCR were quantified with ImageJ software. The results were statistically analyzed using the Student’s *t*-test and a one-way analysis of variance (ANOVA) with Tukey’s post hoc test using the GraphPad Prism (GraphPad Software, San Diego, CA, USA). Values are expressed as means ± standard error of the mean (SEM). The minimum significance level was set at a *p*-value = 0.05. All experiments, except for the animal studies, were independently performed at least three times.

## 3. Results

### 3.1. Effects of Compounds from P. lactiflora on LIF Expression and In Silico ADME Properties

First, we tested six major compounds from *P. lactiflora* [12], and their molecular structures are presented in Figure 1A. Ishikawa cells were treated with these six compounds. To evaluate their potential enhancement of endometrial receptivity, we compared the expression levels of LIF. Our results showed that methyl gallate, paeoniflorin, and paeonol increased LIF protein expression levels (Figure 1B). Second, we investigated the pharmacokinetic parameters of *P. lactiflora* on the basis of the network pharmacology using TCMSP.

Twenty-nine compounds satisfied both the OB ≥30% and the DL ≥0.18 criteria, and, among the previously reported six components of *P. lactiflora*, only (+)-catechin and paeoniflorin and its nine derivative compounds satisfied these conditions (Table 1 and Appendix A). In particular, gallic acid, methyl gallate, and paeonol did not satisfy the OB and DL screening criteria (Appendix A). In addition, (−)-catechin and (+)-catechin hydrate were not included in the list obtained from the TCMSP analysis. According to the data from TCMSP-based in silico ADME and Western blot analysis, paeoniflorin might be an active component of *P. lactiflora*.

### 3.2. Effect of Paeoniflorin on Endometrial Receptivity of Implantation-Depleted Mouse Model

The previously reported RU-induced implantation depletion model [12,30] was used to evaluate the in vivo effect of paeoniflorin on embryo implantation (Figure 2A). The results showed that the number of implanted embryos in the paeoniflorin + RU group (8.17 ± 2.99) was significantly higher than that in the RU group (0.17 ± 0.41). Although the number of embryos in the paeoniflorin (8.50 ± 0.84) and paeoniflorin + RU groups were slightly higher than those in the control group (7.50 ± 1.38), there were no statistically significant differences (Figure 2B,C). The histological observations showed that there were no fetal regression sites in the paeoniflorin-treated groups. In addition, the dose of paeoniflorin used in the animal study did not induce any toxicities on liver or kidney functions according to the analysis of the aspartate transaminase (AST), alanine transaminase (ALT), and BUN (blood urea ni-trogen) levels (Appendix A). These findings indicated that paeoniflorin prevented the implantation defect induced by RU treatment.

### 3.3. Effect of Paeoniflorin on Adhesion between Trophoblast and Endometrial Cells

In this study, paeoniflorin did not show significant cytotoxicity on endometrial Ishikawa cells at concentrations up to 500 μM (Figure 3A). To evaluate the effect of paeoniflorin on endometrial receptivity, adhesion assays were performed with endometrial Ishikawa and fluorescently labeled trophoblastic JAr cells. Ishikawa cells treated with paeoniflorin (50 μM) attached more markedly to JAr cells than the untreated control cells (2.32 ± 0.23-fold, Figure 3B,C).

### 3.4. Role of LIF Expression on Paeoniflorin-Stimulated Cell–Cell Adhesion

We next evaluated the effect of LIF on paeoniflorin-stimulated adhesion between endometrial cells and trophoblasts. The results demonstrated that paeoniflorin increased the expression of both messenger RNA (mRNA) and protein expression levels of LIF in a dose-dependent manner (Figure 4A). In addition, to elucidate whether LIF mediates paeoniflorin-stimulated adhesion between the endometrium and trophoblast, LIF expression was depleted using siRNA. When the expression of LIF was abolished, a significantly lower number of JAr cells attached to the paeoniflorin-treated Ishikawa cells (1 ± 0.12-fold) than to the paeoniflorin-treated control cells (1.81 ± 0.21-fold, Figure 4B,C). These results suggest that LIF is a crucial mediator of paeoniflorin-stimulated endometrium–trophoblast interaction.

### 3.5. Validation of LIF as a Target for Human Endometrial Receptivity

To confirm that LIF is critical in human implantation, we analyzed transcriptomic data obtained from the human endometrium (GSE71835 + GSE92324, GSE26787, and GSE4888). We performed gene set enrichment analysis using the National Center for Biotechnology Information (NCBI) Gene Expression Omnibus (GEO) database. According to the Hallmark gene sets, the expression of genes involved in allograft rejection and interleukin (IL)-6/Janus kinase (JAK)/signal transducer and activator of transcription 3 (STAT3) signaling was upregulated more in the normal endometrium than in the endometrium from RIF patients (GSE71835 + GSE92324, and GSE26787). Furthermore, the levels were more upregulated in the mid-secretory phase than they were in the proliferative phase (GSE4888, Figure 5A–C).

To identify the key factors among each gene set, we determined the genes that were common among the three transcriptomes. There were five genes in the overlapping region of each allograft rejection (*LIF*, *CD247*, *TGFB2*, *C2*, and *INHBB*) and the IL-6/JAK/STAT3 signaling (*OSMR*, *PLA2G2A*, *CSF2RA*, *PIM1*, and *SOCS3*) gene set (Figure 5D,E). These common genes were used to perform the protein–protein physical interaction network analysis. As shown in Figure 5F,G, allograft rejection and the IL-6/JAK/STAT3 signaling gene set were merged using IL-6 signal transducer (IL6ST) and Oncosatin M (OSM). This result suggests that the LIF and OSM pathways could be key targets in the regulation of human endometrial receptivity.

## 4. Discussion

Unexplained recurrent implantation failure is a major clinical unmet need in the treatment of female infertility, especially in the embryo transferring technique [31]. Thus, to improve the receptivity of the endometrium, many researchers have attempted to identify novel effective drugs or technologies such as intentional injury of the local endometrium [4], melatonin [32], prostaglandins [33,34], growth factors [35,36], and chemical drugs [37,38]. Several herbal drugs or their constituents have also been studied as potent candidates for the treatment of implantation failure [38,39,40,41,42]. Furthermore, we also reported that several herbal medicines and natural products enhance embryo implantation through an LIF-mediated mechanism [11,12,13,14].

Although the effect of *P. lactiflora* on embryo implantation was examined in our previous study [12], the precise active component mediating its endometrial receptivity-enhancing effects was not elucidated. This study aimed to identify the active compound in *P. lactiflora*. Paeoniflorin increased LIF expression and the attachment of trophoblasts to the endometrium in the cell adhesion assay at nontoxic concentrations. Paeoniflorin improved the embryo implantation rate in the RU-induced murine implantation failure model. In addition, paeoniflorin satisfied the screening criteria for the examination of ADME processes. Although paeonol and methyl gallate induced LIF expression, they did not satisfy both the OB ≥30% and the DL ≥0.18 requirements for in silico ADME analysis. Additionally, paeonol was reported to exhibit contraceptive activity in mice [43] and, therefore, we suggest that paeoniflorin is a major active compound mediating the improved endometrial receptivity.

LIF has been generally accepted to play a key role in regulating uterine receptivity. However, a controversial report suggests that LIF alone is not sufficient for assessing the implantation potential in humans [44]. *LIF* gene mutation does not frequently occur in infertile women [45], and recombinant human LIF did not improve implantation rates after in vitro fertilization (IVF) [46]. On the other hand, recent mRNA sequencing studies revealed that LIF/STAT3 signaling is reduced in the endometrium of RIF patients [47]. However, the female volunteers who provided endometrial tissues were all Korean.

To estimate the correlation between LIF and endometrial receptivity in humans, we analyzed several NCBI GEO data of human endometrial tissues obtained from diverse countries including India, France, and the United States of America (USA) (GSE71835 + GSE92324, GSE26787, and GSE4888). According to the GSEA, the genes involved in allograft rejection and IL-6/JAK/STAT3 signaling were related to high endometrial receptivity in the three NCBI GEO datasets. The allograft rejection pathway is the adaptive immune response for allotransplantation, which is the transplantation of a tissue or organ into an individual from a genetically different donor of the same species [48], and it is mediated by diverse cytokines [49].

Among the IL-6 family of cytokines, IL-6, LIF, and OSM are regarded as critical regulators [50]. Their receptors consist of two subunits containing at least one gp130 molecule, which is also known as IL6ST [51]. Therefore, all IL-6 family cytokines can trigger the IL-6/JAK/STAT3 signal through the gp130 subunit of their receptors [52]. This indicates that allograft rejection and IL-6/JAK/STAT3 signaling are closely related. Our GSEA results showed that these two gene sets were upregulated in the normal endometrium, suggesting that the LIF/IL6ST/JAK/STAT3 signaling axis might be critical for human endometrial receptivity.

Although we demonstrated that paeoniflorin enhanced endometrial receptivity in vivo and in vitro by expressing LIF, the detailed molecular mechanisms were not fully examined. First, the mechanism via which paeoniflorin induces LIF expression was not established in this study. LIF is regulated by diverse pathways, such as p53, estrogen receptor α, JAK/STAT3, protein kinase B (AKT), extracellular signal-regulated kinase ½ (ERK1/2), and mammalian target of rapamycin (mTOR) signaling [7,53,54]. To elucidate the mechanism, the precise molecular target of paeoniflorin mediating LIF induction should be identified. Several previous studies revealed the molecular targets of paeoniflorin, such as adenosine A1 receptor, liver X receptor, *N*-methyl-d-aspartate receptor, and cannabinoid receptor 2 [55,56,57]. However, no previous study revealed the involvement of these molecular targets in LIF expression. Therefore, further investigations into which targets are related to LIF induction are required. Second, a previous study showed that adhesion molecules, especially integrins, such as integrin α1, α4, αV, β1, and β5, mediate the attachment between the embryo and endometrium [58]. The expression of integrins has also been reported to be diminished in the endometrium of infertile patients [59] and affected by LIF [8,9,10,12]. In addition, OSM, which is significantly upregulated in normal endometrial transcriptomes, has also been reported to promote embryo implantation in mice, and its expression is affected by LIF [60]. Thus, to precisely understand the mechanism via which paeoniflorin enhances endometrial receptivity, further extensive studies of these molecular targets are needed, including OSM and adhesion molecules.

To the best of our knowledge, there is no previous report on the toxicity or safety of paeoniflorin, especially its genotoxicity or reproductive toxicity. Several previous studies have shown that paeoniflorin reduced the growth of various tumor cell lines through cell-cycle arrest [61,62,63,64,65]. However, many studies have reported the preventive effects of paeoniflorin against renal injury [66], neurotoxicity [67], and hepatotoxicity [68]. In this study, paeoniflorin did not show cytotoxicity against human endometrial adenoma Ishikawa cells. An in vivo study also showed that paeoniflorin does not significantly affect liver and kidney functions. In addition, histological examinations showed no significant evidence of fetal regression in the paeoniflorin-treated groups. These results collectively suggest that paeoniflorin did not exhibit severe toxicity on the embryo or maternal mice at the dose used in this study.

## 5. Conclusions

In conclusion, we showed that paeoniflorin, a major component of *P. lactiflora*, enhanced endometrial receptivity in both in vitro and in vivo models through the expression of LIF. Although the clinical efficacy and safety of paeoniflorin should be confirmed by further extensive experiments, we suggest that paeoniflorin is a potent, potential candidate for improving the receptivity of the endometrium and, consequently, treating implantation failure.

## Figures and Tables

**Figure 1 biomolecules-11-00439-f001:**
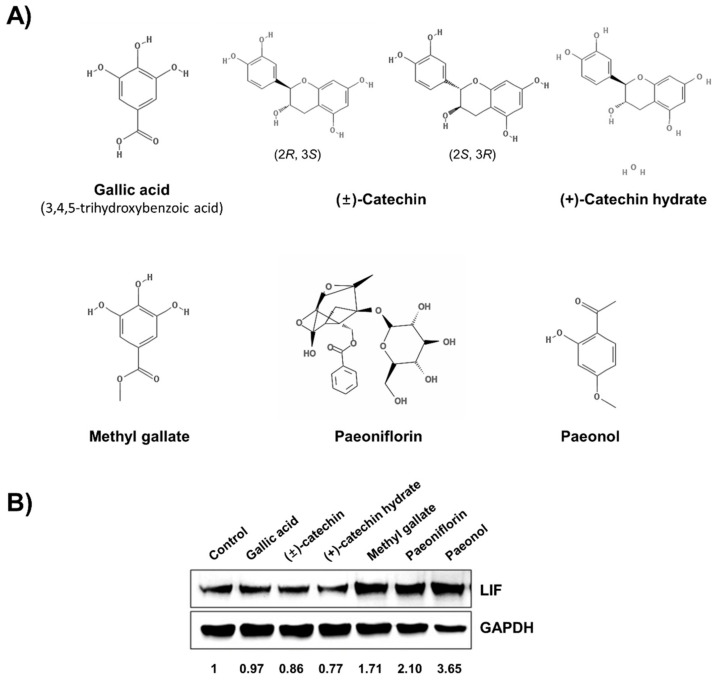
Paeoniflorin is the most active component of *Paeonia lactiflora* in improving endometrial receptivity. (**A**) Structures of major components of *P. lactiflora*. (**B**) Ishikawa cells were treated with indicated six compounds for 24 h and then harvested for Western blot analysis. The expression level of leukemia inhibitory factor (LIF) was used for screening.

**Figure 2 biomolecules-11-00439-f002:**
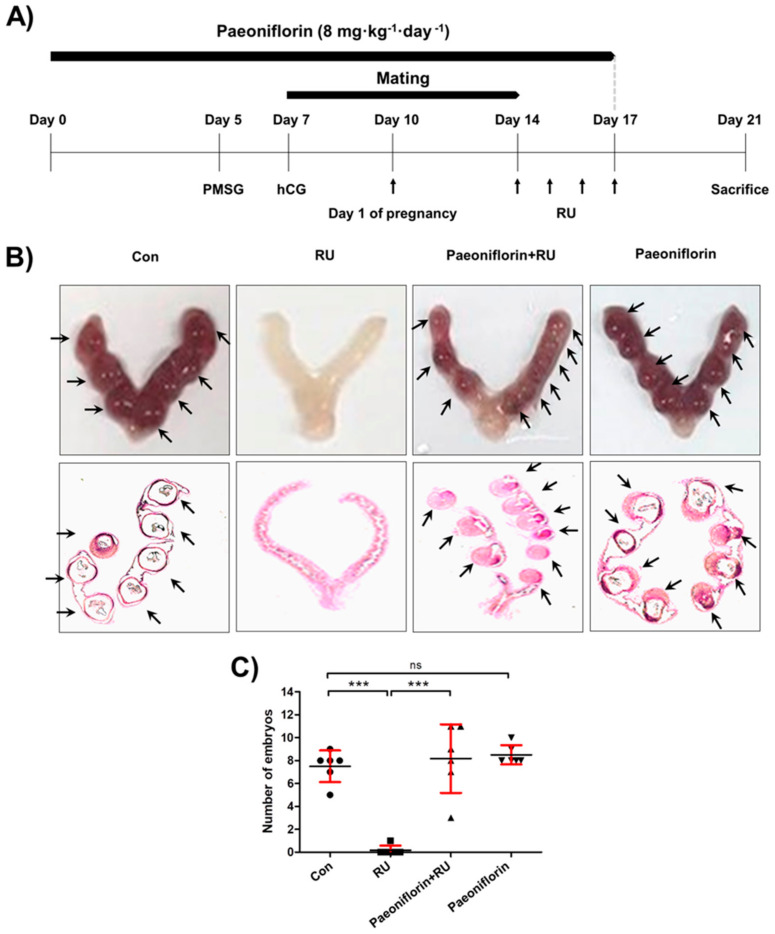
Paeoniflorin enhanced embryo implantation in mifepristone (RU486)-induced implantation failure mouse model. (**A**) Female mice were administered paeoniflorin (8 mg·kg^−1^·day^−1^) for 17 days. Seven days after paeoniflorin administration, female mice were mated with males. On day 4 from pregnancy, evidenced by vaginal plugs, female mice were daily treated with RU (4 mg·kg^−1^·day^−1^) by subcutaneous injection for 4 days. (**B**) Seven days after RU injections, mice were euthanized and the uteri were excised. Representative image of an embryo-implanted uterus is shown. Uterine tissue sections were stained with hematoxylin and eosin (H&E) and histologically analyzed. (**C**) The number of embryo implantation sites was counted and is expressed as the mean ± standard error of the mean (SEM); *** *p* < 0.001 compared to other groups; ns, no significance.

**Figure 3 biomolecules-11-00439-f003:**
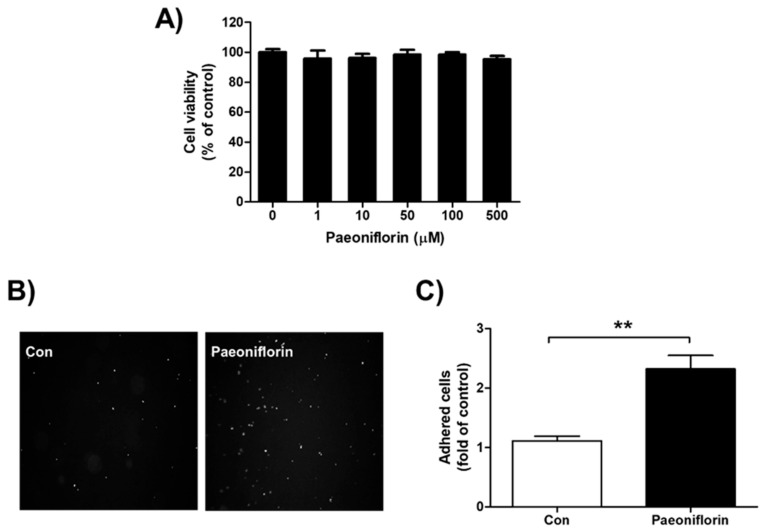
Cytotoxicity and effect of paeoniflorin on adhesion between endometrium and trophoblast. (**A**) Ishikawa cells were treated with 0, 1, 10, 50, 100, and 500 μM paeoniflorin for 24 h. Viable cells were stained with 3-(4,5-dimethylthiazol-2-yl)-2,5-diphenyltetrazolium bromide (MTT) and the produced formazan was measured. Data were calculated as a percentage of control and are shown as the mean ± standard error of the mean (SEM). (**B**) Endometrial Ishikawa cells were treated with 50 μM paeoniflorin for 48 h. Fluorescently labeled trophoblastic JAr cells were added to confluent Ishikawa cells. After incubation and washing, images of attached JAr cells were randomly captured, and representative images are shown. (**C**) The number of attached JAr cells was counted and calculated as a fold-change compared to control, and the results are shown as the mean ± standard error of the mean (SEM); ** *p* < 0.01 compared with control.

**Figure 4 biomolecules-11-00439-f004:**
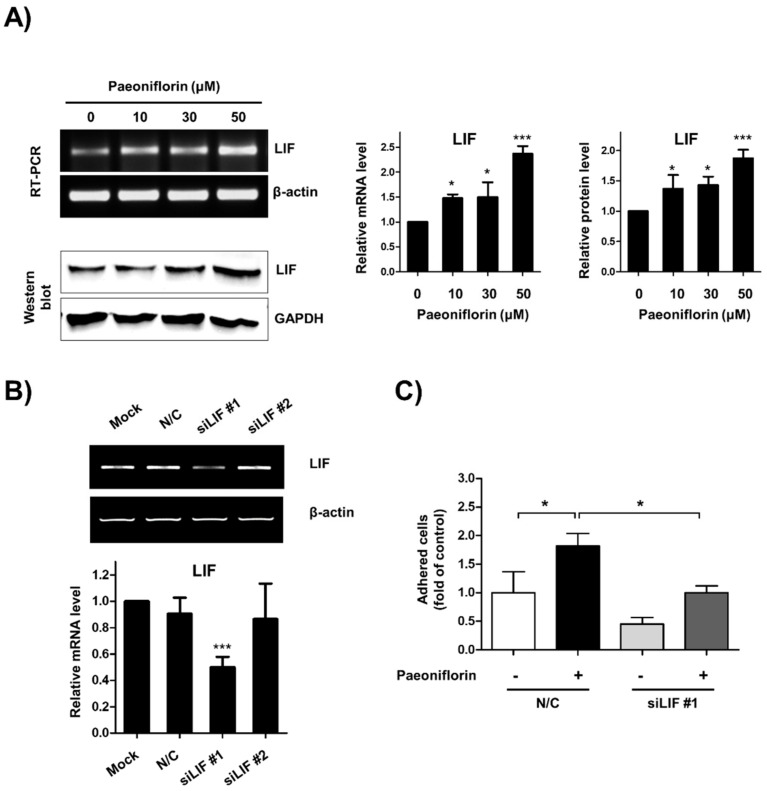
Paeoniflorin enhanced endometrial receptivity through expression of leukemia inhibitory factor (LIF). (**A**) Ishikawa cells were treated with 0, 10, 30, and 50 μM paeoniflorin for 24 h. LIF expression was determined using reverse transcription polymerase chain reaction (RT-PCR) and Western blot analysis. β-Actin and glyceraldehyde 3-phosphate dehydrogenase (GAPDH) were internal controls. Relative mRNA and protein levels of LIF are shown as the mean ± standard error of the mean (SEM); * *p* < 0.1 and *** *p* < 0.001 compared to control (0 μM). (**B**) Ishikawa cells were treated with carrier only (mock), negative control small interfering RNA (siRNA, N/C), and siRNA against LIF. After 24 h incubation, LIF expression was determined using RT-PCR. The relative mRNA level of LIF is shown as the mean ± standard error of the mean (SEM); *** *p* < 0.001 compared to mock. (**C**) Ishikawa cells were transfected with N/C or siRNA against LIF. These cells were treated with or whithout 50 μM paeoniflorin for 48 h (labelled with +, -). Fluorescently labeled JAr cells were added to confluent Ishikawa cells. After incubation and washing, the number of attached JAr cells was counted and calculated as a fold-change compared to control, and the results are shown as the mean ± standard error of the mean (SEM); * *p* < 0.01 compared to each group.

**Figure 5 biomolecules-11-00439-f005:**
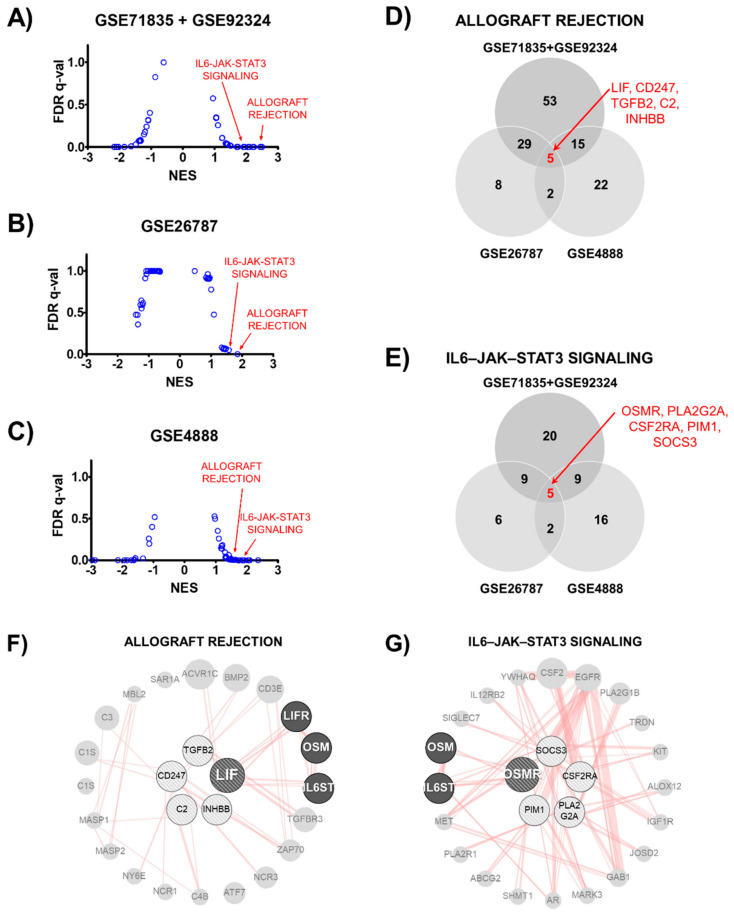
Gene expression profiles in human endometrium according to Hallmark gene sets. (**A**–**C**) Dot-plot graph showing ranked gene sets using normalized enrichment score (NES). Allograft rejection and interleukin-6/Janus kinase/signal transducer and activator of transcription 3 (IL-6-JAK-STAT3) signaling gene sets had NES >1.5 in normal (GSE71835 + GSE92324 and GSE26787) and secretory (GSE4888) human endometrium. (**D**,**E**) Venn diagram shows number of common genes among three transcriptomes in allograft rejection or IL-6/JAK/STAT3 signaling gene set. Five genes were in the overlapping region of each gene set. (**F**,**G**) Protein-protein interaction network of overlapping genes. Overlapping genes are located in center of a circular network. The line indicates a physical interaction between connected proteins.

**Table 1 biomolecules-11-00439-t001:** Twenty-nine components of *Paeonia lactiflora* satisfied oral bioavailability (OB) ≥30% and drug likeness (DL) ≥0.18 obtained from Traditional Chinese Medicine Systems Pharmacology (TCMSP).

Molecule Name	MW	OB (%)	DL	AlogP	Hdon	Hacc	Caco-2
Paeoniflorgenone *	318.35	87.59	0.37	0.79	1	6	−0.09
Paeoniflorin_qt *	318.35	68.18	0.4	0.46	2	6	−0.34
Paeoniflorigenone *	318.35	65.33	0.37	0.79	1	6	−0.13
1-*O*-*β*-d-Glucopyranosylpaeonisuffrone_qt	332.38	65.08	0.35	0.51	1	6	−0.05
Evofolin B	318.35	64.74	0.22	2.07	3	6	0
9-Ethyl-neo-paeoniaflorin A_qt *	334.4	64.42	0.3	1.48	1	6	−0.01
(2*R*,3*R*)-4-Methoxyl-distylin	318.3	59.98	0.3	1.89	4	7	0.17
4-Ethyl-paeoniflorin_qt *	332.38	56.87	0.44	1.02	1	6	−0.17
4-*O*-Methyl-paeoniflorin_qt *	332.38	56.7	0.43	0.87	1	6	0.4
(+)-Catechin	290.29	54.83	0.24	1.92	5	6	−0.03
Paeoniflorin *	480.51	53.87	0.79	−1.28	5	11	−1.47
Lactiflorin	462.49	49.12	0.8	−0.57	3	10	−1.13
Albiflorin_qt	318.35	48.7	0.33	0.42	2	6	−0.38
Stigmasterol	412.77	43.83	0.76	7.64	1	1	1.44
Ellagic acid	302.2	43.06	0.43	1.48	4	8	−0.44
Spinasterol	412.77	42.98	0.76	7.64	1	1	1.44
Baicalin	446.39	40.12	0.75	0.64	6	11	−0.85
Campest-5-en-3*β*-ol	400.76	37.58	0.71	7.63	1	1	1.32
Stigmast-7-en-3-ol	414.79	37.42	0.75	8.08	1	1	1.32
*β*-Sitosterol	414.79	36.91	0.75	8.08	1	1	1.32
Sitosterol	414.79	36.91	0.75	8.08	1	1	1.32
1-*O*-*β*-D-Glucopyranosyl-8-*O*-benzoylpaeonisuffrone_qt *	302.35	36.01	0.3	0.44	1	5	−0.03
Baicalein	270.25	33.52	0.21	2.33	3	5	0.63
Ethyl oleate (NF)	310.58	32.4	0.19	7.44	0	2	1.4
8-Debenzoylpaeonidanin	390.43	31.74	0.45	−3.28	5	10	−1.56
Benzoyl paeoniflorin *	584.62	31.14	0.54	0.76	4	12	−1.35
Isobenzoylpaeoniflorin *	584.62	31.14	0.54	0.76	4	12	−0.85
Albiflorin	480.51	30.25	0.77	−1.33	5	11	−1.52
(1*S*,2*S*,4*R*)-trans-2-Hydroxy-1,8-cineole-*β*-D-glucopyranoside	332.44	30.25	0.27	−0.57	4	7	−0.77

Note: * paeoniflorin and its derivatives. MW, molecular weight; OB, oral bioavailability; DL, drug likeness; AlogP, octanol–water partition coefficient log P; Hdon, hydrogen donor; Hacc, hydrogen bond acceptor; Caco-2, Caco-2 permeability.

## Data Availability

Publicly available datasets were analyzed in this study. These data can be found as https://www.ncbi.nlm.nih.gov/ (accessed on 15 March 2021) GSE71835, GSE92324, and GSE4888.

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
