# Peer review of "Paeoniflorin Enhances Endometrial Receptivity through Leukemia Inhibitory Factor"

_biomolecules, 2021, doi:10.3390/biom11030439_

Round 1

Reviewer 1 Report

The study was aimed at testing the effcacy of Paeoniflorin  in enhancing implantation in a mose model. In a previous paper the authors showed that Paeonia lactiflora increases the adhesion of throphoblast cells to the endometrium. In this paper they show that Paeoniflorin is the active compound of Paeonia lactiflora. The study is well conducted and results are sound.

Author Response

Thank you for considering our manuscript.

Reviewer 2 Report

The manuscript by Hye-Rin Park demonstrated, both in vitro and in vivo models, that paeoniflorin is able to enhance endometrial receptivity through expression of LIF. The results are clear and well presented, but in some cases one single experiment is not sufficient to draw a conclusion. The results of western blotting reported in Figure 1B must be repeated at least three times, the density of bands must be measured and mean values, with the standard deviation and the statistical analysis should be reported. The same for data reported in Figure 4A and 4B. In some sections, the English language should be improved, see for example Lines 222-223. One important paper on the argument (J Tradit Chin Med. 2019 Feb;39(1):15-25) has not been cited, I wonder why?

Author Response

(The authors gave the same response as above.)

Reviewer 3 Report

Park et al., showed that paeoniflorin,a major active compound of P. lactiflora,  significantly improved the embryo implantation rate in a murine model of RU486-induced implantation failure. In addition, paeoniflorin increased the adhesion of human trophectoderm-derived JAr cells to endometrial Ishikawa cells through the expression of LIF in vitro. This manuscript was accepted after major revision.

Introduction:

Authors should include the medicinal use of Paeonia lactiflora and focus and on its Chinese  traditional usage.

Experimental parts:

-Authors should include references in the expermintal parts, for example: MTT, bioinformatics, etc…

-L113, Western blot analysis, there is no details about the experimental and antibody dilution.

-L115. Total proteins were isolated from each sample using…………….., which samples?

Results: L-171, authors test six although ther are 7 compounds, although there are enantiomers. If authors check six compounds, accordingly, they should draw catechin as racemic in Figure 1.

-Why authors donot check -catechin and +catechin separately, especially they are commercially available.

- Table 1: is it 29 or 30 compounds listed on the table, please check?

- Why authors select paeoniflorin not paeonol, especially the last one showed upregulation of LIF protein, especially both are major metabolites!!!!!!!

-All stereo (R,S, Z, Alpha or Beta(B) should be italic.

-I suggest that authors could improve this work through molecular docking  insilico studies.

Author Response

(The authors gave the same response as above.)

Round 2

Reviewer 3 Report

This manuscript is accepted